# CMoB: Modality Valuation via Causal Effect for Balanced Multimodal Learning

**Jun Wang[1], Fuyuan Cao[1,2],\* Zhixin Xue[1], Xingwang Zhao[1], Jiye Liang[1]**
[1]School of Computer and Information Technology, Key Laboratory of Computational
Intelligence and Chinese Information Processing of Ministry of Education,
Shanxi University, Taiyuan, China
[2]Shanxi Taihang Laboratory, Taiyuan, China
jwang8532@gmail.com, cfy@sxu.edu.cn, xuezhixin@sxu.edu.cn,
zhaoxw@sxu.edu.cn, ljy@sxu.edu.cn

## Abstract

Existing early and late fusion frameworks in multimodal learning are confronted with the fundamental challenge of modality imbalance, wherein disparities in representational capacities induce inter-modal competition during training. Current research methodologies primarily rely on modality-level contribution assessments to measure gaps in representational capabilities and enhance poorly learned modalities, overlooking the dynamic variations of modality contributions across individual samples. To address this, we propose a **C**ausal-aware **Mo**dality valuation approach for **B**alanced multimodal learning (CMoB). We define a benefit function based on Shannon's theory of informational uncertainty to evaluate the changes in the importance of samples across different stages of multimodal training. Inspired by human cognitive science, we propose a causal-aware modality contribution quantification method from a causal perspective to capture fine-grained changes in modality contribution degrees within samples. In the iterative training of multimodal learning, we develop targeted modal enhancement strategies that dynamically select and optimize modalities based on real-time evaluation of their contribution variations across training samples. Our method enhances the discriminative ability of key modalities and the learning capacity of weak modalities while achieving fine-grained balance in multimodal learning. Extensive experiments on benchmark multimodal datasets and multimodal frameworks demonstrate the superiority of our CMoB approach for balanced multimodal learning.

## 1 Introduction

Humans construct multi-dimensional perception through multiple sensory modalities like vision, touch, hearing, and smell, processing information hierarchically to understand the real world [1, 2, 3]. This biological cognitive mechanism has inspired the development of multimodal learning paradigms in the field of machine learning. These paradigms involve constructing collaborative learning frameworks across heterogeneous modal representation spaces (e.g., text, images, audio, and video) to simulate the cognitive process of human cross-modal information fusion [4]. From a cognitive neuroscience perspective [5, 6, 7], this paradigm aligns with the neural mechanisms of the human brain's multisensory integration, where different sensory cortices enable collaborative learning across heterogeneous modalities through synaptic plasticity.

In recent years, although extensive research on multimodal learning has made significant progress, some studies have found that most existing multimodal models encounter the challenge of modality

---

\*Corresponding author.

39th Conference on Neural Information Processing Systems (NeurIPS 2025).

collapse during joint training. This is because deep neural networks tend to prioritize modalities that are easy to learn while neglecting other modalities, thus failing to effectively integrate heterogeneous cross-modal information [8, 9, 10]. This issue prevents the model from fully leveraging the complementary information across all modalities. Consequently, this information loss results in the performance of unimodal in joint multimodal training falling far short of their intrinsic performance ceilings. Researchers define this phenomenon as multimodal imbalance learning [11, 12, 13, 14, 15]. The primary cause of this phenomenon stems from the greedy nature of deep neural networks: these networks tend to rely on high-quality modality that is highly relevant to the target task, thereby inhibiting the optimization of other modalities. In the joint training process of multimodal learning, this greediness makes the modality dominant in the joint training, inhibiting the learning of other modalities and generating a situation of modal competition. Researchers define this phenomenon as multimodal imbalance learning [16, 17].

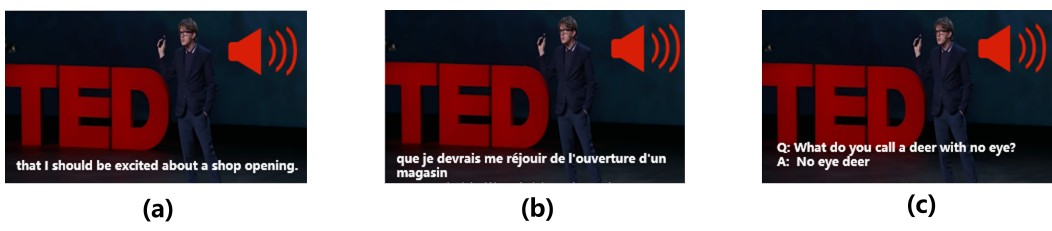

(a)        (b)        (c)

Figure 1: A TED talk with audio, video, and text modalities.

Recent research has focused on mitigating inter-modal discrepancies and enhance modality-specific data utilization during training to address the above challenge for improved multimodal model efficacy. Several methods have been proposed to identify and improve the training of poorly learned modalities by controlling gradient variations, modality contribution and incorporating loss functions [15, 18, 19, 20]. Gradient-based methods for assessing modality contribution typically rely on the assumption that larger gradients indicate higher modal importance. However, gradient values are susceptible to fluctuations from model parameters and training data, leading to unstable evaluations. Even though all modalities of a sample express the same concept (i.e., share identical label information), the amount of information related to the target object or target event varies across different modalities, which leads to differences in the data quality of individual modalities. These methods are only limited to modality-level analysis and fail to capture contribution differences at the sample level. In many real-world datasets, not all samples of image modal data, video modal data are more informative than text modal data. Take a TED talk video as an example (as shown in Figure 1): for video comprehension, the speech stream (audio information) and subtitle stream (text information) of a TED talk convey consistent core information. Viewers typically adopt intuitive, information-rich audio information as their primary source, parsing the speaker's speech stream to understand the meaning of the conveyed content. But when the target language suddenly exceeds the receiver's comprehension scope ((e.g., during a non-native language talk), the subtitle text replaces the auditory channel and becomes the core cognitive pathway for understanding the expression. Assessing the contribution degree of modalities at the modal level fails to reflect the true nature of the data. And although current encoders are effective in extracting multimodal features, the lack of interpretability [21, 22, 23] of the depth model makes it difficult to observe the role of each modality in the final prediction and to adjust unimodal training accordingly.

Inspired by the multimodal cognitive mechanisms of the human nervous system, we address the above problem by analyzing the learning process of the human brain's integrated processing of heterogeneous multimodal data [5, 7, 24, 25, 26]. We illustrate this concretely using a TED talk video example, as shown in Figure 1(d). When the speaker utters "no eye deer"—a homophonic phrase identical in pronunciation to "no idea"—viewers relying solely on the audio stream may struggle to grasp the intended meaning. In such cases, the cognitive system leverages instantaneous information entropy [27] to dynamically integrate subtitle data for assisted comprehension. During this process, the original cognitive system first extracts discriminative causal features [28] from audio data and based on experience to obtain the information entropy value of this modality. It then dynamically selects text modality data to assist in comprehension, thereby continuously refining the cognitive system's understanding of these modalities. We analyze this process and find that it has two core characteristics: (1) modality contribution valuation based on causality. By acquiring discriminative feature information and then evaluating modality contribution according to its information entropy,

rather than simple signal quality comparison; (2) granular adjustment of the sample level. Even if a modality is overall reliable in the temporal dimension, the cognitive system will still implement cross-modality switching when an information bottleneck occurs in a single sample, and re-optimize its cognition of the modality. Inspired by this cognitive mechanism, in this paper we revisit the process of multimodal learning at the sample level from the perspective of causal learning. Causal learning aims to explore the essential causal relationship between things and reveal the real generation mechanism inside the data. We evaluate the degree of modality contribution by measuring the causal effect between samples, distinguishing the role of each modality in the final prediction and identifying low contributing modal samples. During the model training process, we enhance and optimize the low contributing modal samples to selectively improve the learning direction of the multimodal model to alleviate the modality imbalance problem and improve the performance and interpretability of multimodal learning.

We propose a causal-aware modality valuation method to evaluate the sample-level modality contribution in multimodal training. We design an optimization strategy for modality selection at the sample-level according to the contribution degree of each modality in order to mitigate the modality imbalance problem. In our experiments, we validate the effectiveness of our method by comparison and ablation experiments on publicly available data. In summary, our main contributions are as follows: (1) We evaluate the importance of sample in multimodal learning by means of a benefit function designed by information uncertainty theory. (2) We propose methods to quantify the degree of contribution of each modality from a causal perspective and represent the contribution of modalities in terms at the sample-level. (3) We propose a modality balancing approach from a data perspective to improve the performance of multimodal learning by optimizing the selection of weak modalities from the sample level in terms of modality contribution. (4) We validate the effectiveness and benefits of our approach by conducting extensive experiments on three publicly available datasets for different modalities compared to existing work.

## 2   Related work

### 2.1   Imbalance multimodal learning

The development of multimodal learning has been a great success, but many recent studies have found that the unimodal capabilities are not fully exploited in multimodal learning [1, 29, 30, 31]. Due to the imbalance in modality contributions during training, some unimodal capabilities are inhibited by the dominant modality and are unable to exert their upper limits [2, 8, 10, 17, 11, 15]. In some special cases, the performance of multimodal algorithms can even be lower than the performance of unimodal algorithms. Some researchers mitigate modality imbalance by analyzing the causes of modality imbalance and by modifying the process of forward propagation of model inference. For example, OPM algorithm [14]adopts a dynamic feature discarding approach for modality during multimodal training. OGM [17]achieves balanced multimodal learning by suppressing the dominant modality through an adaptive gradient adjustment strategy. PMR [12] accelerates the learning of weak modalities through category prototyping. GBlending [32]and MMpareto [16]reduce gradient conflicts during training by introducing unimodal assisted learning to control the direction of gradient update. Based on the above analysis, although these methods alleviate the modality imbalance to a certain extent, their approaches measure the degree of modality contribution through the modality-level and lack interpretability. Many scholars seek to investigate this issue from a sample-level perspective. SMLS [33] employs KL divergence to align the factual contribution distribution with the utopia contribution distribution. PDF [34]aims to reduce reliance on low-quality modalities via Relative Calibration (RC). The shapley value-based method [8], which enhances weak modalities by calculating combinations of all possible subsets. In this paper, we propose an interpretable method to quantify the degree of contribution of each modality in multimodal training. We characterize the degree of modality contribution in terms at the sample-level to mitigate the modality imbalance issue.

### 2.2   Causal learning

Causal learning aims to discover causal relationships from data and utilize these relationships to make interventions, predictions and decisions [35]. Causal learning focuses on identifying causal relationships, which is different from traditional machine learning that only discovers correlations between variables. Causality strictly distinguishes between "cause" and "effect" variables, and

plays an irreplaceable role in revealing the mechanism of occurrence and guiding intervention behavior [36, 37, 38]. Causal inference has contributed to the development of artificial intelligence [26, 39, 40, 41] due to its ability to eliminate the harmful bias of confounders and to discover causal relationships among multiple variables. Causal effect are a central goal of causal inference, aiming to quantify the impact of interventions on outcomes by comparing the outcomes of treatment and control groups. Specifically, causal effect reveal causal relationships between variables by analyzing differences between potential outcomes across treatment conditions [28]. The central question is to answer: for a given individual or group, how do outcomes change when a certain intervention is applied compared to not applying that intervention? In this paper, we revisit the challenge of modality imbalance in multimodal joint training from a causal learning perspective.

## 3 Methodology

This section presents our method. First, we formulate the multimodal learning problem and its representation paradigm. Then, we propose an information-theoretic uncertainty measure for sample significance valuation and develop a causal-aware algorithm to quantify modalities contribution at the sample level. Finally, we propose a modality rebalancing approach from a data perspective to improve multimodal learning by optimizing the selection of weak modalities from the sample-level modality contribution.

### 3.1 Preliminary

With any loss of generality, We formalize the general formalism for expressing multimodal learning. Given a dataset $\mathcal{D} = \{(x_i^m, y_i^m) \mid i \in \{1, \cdots, N'\}, m \in \{1, \cdots, M\}\}$ is a finite and nonempty set for all modalities with $N'$ data samples and $M$ modalities. Each sample $x$ has $M$ modality, and $y$ is the label of $x$ can refer to a class, an answering, etc. $x_i = \{x_i^1, x_i^2, \cdots, x_i^M\}$, $y_i \in \{1, 2, \ldots, K\}$ denotes the corresponding class label from $K$ classes. For ease of expression, we give examples with two modalities where $v$ and $a$ refer to the video and audio modalities, respectively. It is worth noting that our method is not limited to two modalities and can be extended to more modalities.

For multimodal joint training methods, we use the mainstream $M$ modality-specific encoder to extract features from the original space. We use $h(\cdot)$ to define the feature extraction of the multimodal model. For any sample, the feature extraction process can be formalized as:

$$z_i^m = h(\theta^m, x_i^m),$$

where $z$ denotes the $d$-dimension feature vector $z_i^m \in \mathbb{R}^d$, $\theta^m$ denote the parameters of modality-specific encoder $\Theta^m(\theta^m, \cdot)$.

We obtain a joint representation of modality-specific features by fusing them through a fusion function $f(\cdot)$. Then, we use $\hat{F}(\cdot)$ to denotes the output function of the multimodal model, which maps vectors into $\mathbb{R}^K$, This procedure can be formally formulated as:

$$Z = f(z_i^1, z_i^2, \cdots, z_i^M), \ Z \in \mathbb{R}^D, \quad \hat{F}(W, \theta, x, b) = W \cdot Z + b,$$

where $D$ denotes the dimension of the joint feature representation, $W \in \mathbb{R}^{K \times D}$ denotes the weight of the last layer of the forward propagation process, $K$ denotes the number of categories in the task, and $b$ is the bias term, $b \in \mathbb{R}^k$.

Finally, the cross-entropy loss is used for validation and the joint objective loss for multimodal learning can be formalized as:

$$L_{\text{cross-entropy}} = \frac{1}{N'} \sum_{i=1}^{N'} y_i \cdot \log(\text{softmax}(\widehat{F})).$$

When updating the gradient for model backpropagation, $W^m$ and the parameters of modality-specific encoder $\Theta^m$ are updated as:

$$W_{t+1}^m = W_t^m - \eta \frac{1}{N'} \sum_{i=1}^{N'} \frac{\partial L_{\text{cross-entropy}}}{\partial \hat{F}(x_i)} \cdot \Theta_i^m, \quad \theta_{t+1}^m = \theta_t^m - \eta \frac{1}{N'} \sum_{i=1}^{N'} \frac{\partial L_{\text{cross-entropy}}}{\partial \hat{F}(x_i)} \cdot \frac{\partial (W_t^m \cdot \Theta_i^m)}{\partial \theta_t^m},$$

$$\frac{\partial L_{\text{cross-entropy}}}{\partial \hat{F}(x_i)} = \frac{e^{\sum_{m=1}^{N'} W^m \cdot \Theta_i^m + b_{\hat{y}_i}}}{\sum_{k=1}^{K} e^{\sum_{m=1}^{N'} W^m \cdot \Theta_i^m + b_k}} - \mathbf{1}_{\hat{y}_i = y_i},$$

where $\eta$ is the learning rate, $\hat{y}_i$ as the classification result of $x_i$. The phenomenon of modality imbalance can be formalized as: when one modality prediction has better outcome, its contribution $W^m \cdot \Theta_i^m$ dominates the logits output $\hat{F}(x_i, \cdot)$. This reduces the magnitude of $\frac{\partial L_{\text{cross-entropy}}}{\partial \hat{F}(x_i)}$, as the loss $L_{\text{cross-entropy}}$ already becomes smaller. The model is also optimized in the direction of smaller loss. Consequently, gradients for updating weaker modalities are suppressed, leading to under-optimized representations for them.

### 3.2 Sample benefit valuation

In multimodal learning, the multimodal model utilizes the different modalities of all samples for learning and inference, thereby enhancing the understanding of the data. For existing tasks of multimodal learning, all modalities are assumed to be predictive of the model [30]. The benefit function we designed is also based on this assumption. We mainly address the modality imbalance phenomenon of early fusion and late fusion in multimodal learning. We take the number of input modalities $M$ as the value of the benefit function in multimodal joint learning when the prediction of the final network learning is matched with the ground truth.

In multimodal learning, the importance of samples changes as the model is trained iteratively. Due to the heterogeneity of modalities, the modality data itself also contains noise, inter-modal redundant information. In the initial stage of training, this may prevent the model from learning the high-level semantic relationship information of different modalities. In real scenarios, the data quality of different modalities of a sample is usually unstable, not all samples will have higher video modality than text modality [8]. In reality, multimodal models rely on different modalities for different samples, rather than depending on the same modality for all samples. Moreover, for different tasks in the same dataset, the amount of information related to different target tasks in the same sample is different, so it is difficult to accurately define the "quality" of modality in each sample.

According to Shannon's theory of information uncertainty [42], "the essence of information is to eliminate uncertainty" brings us new thinking. In other words, adding more modalities to a sample is accompanied by enhancing more information, which will bring less uncertainty to the multimodal model. We can approximate this relationship as follow:

**Proposition 1.** Given a sample has $M$ modalities denoted as $x^{(M)} = \{x^1, x^2, \cdots, x^M\}$, assume that there any two subsets $x^{(B)}, x^{(C)}$ of $x^{(M)}$, and $x^{(B)} \subseteq x^{(C)} \subseteq x^{(M)}$, then for any multimodal classifier $\hat{F}(\cdot)$, it should be guaranteed that

$$\text{Conf}(\hat{F}(x^{(B)})) < \text{Conf}(\hat{F}(x^{(C)})) \leq \text{Conf}(\hat{F}(x^{(M)})).$$

Current encoders and classifiers have achieved significant advancements. For any reliable multimodal classifier, when new modalities are added, the prediction confidence of multimodal joint learning should increase and exhibit a positive correlation with the addition of modalities.

We defined a benefit function to evaluate the importance of the samples in the multimodal model learning process as follow:

$$B(M) = \begin{cases} |M|, & \text{if } \hat{F} = y_i, \text{ Conf}(\hat{F}(x^{(C)})) \leq \text{Conf}((\hat{F}(x^{(M)}), \text{ and } C \leq M, \\ 0, & \text{otherwise.} \end{cases} \tag{1}$$

When the multimodal model's prediction is correct and the confidence after adding new modalities is higher than that before their addition, in this case the benefit of this sample is defined as the number of input modalities.

### 3.3 Causal-aware quantification method

In this subsection, we propose a causal-aware modality contribution quantification method from a causal perspective to capture fine-grained changes in modality contribution within samples. We dynamically evaluate the contribution degrees of different modalities in samples by calculating the changes in causal effects due to intervention modalities.

In the previous subsection, we evaluated the benefit value of the sample to the target task and measured the importance of the sample through the benefit function. Cognitive science research has revealed a unique mechanism in multimodal information processing in which the human modality choice behavior has a significant causal inference property [5]. This characteristic suggests that humans do not simply integrate all available information when faced with complex multimodal environments, but selectively focus on and utilize modality-specific information based on judgments of causality and causal strength.

Intervention is used as a straightforward and effective method to assess the existence of direct causal relationships between events [28, 35, 39]. Inspired by the concept of intervention in causality, we assess the effect of one event (cause) leading to the occurrence of another event (outcome) by means of intervention, and quantify the degree of contribution of modality in the sample. For multimodal deep neural networks, we observe the causal relationship between the variable $t_i$ of the input set $T$ and its corresponding output $H(T)$ through an intervention denoted as $do(t_i = x)$, as shown in Eq.2

$$\phi_H[t_i] = \mathcal{V}(H(T)) - \mathcal{V}(H(T|do(t_i = x))) = \mathcal{V}(H(T)) - \mathcal{V}(H(T')). \tag{2}$$

Here, $H(T)$ denotes control group, $H(T')$ denotes the treatment group. The difference between these two refers to the effect of the intervention. $\phi_H[t_i]$ can be regarded as the ITE(Individual Treatment Effect) of $t_i$ through the function $H(\cdot)$. $\mathcal{V}(\cdot)$ can be any function that valuates the effect of an outcome. $T'$ denotes the intervened outcome.

For each sample which contains M heterogeneous modalities, $S(x_i) = \{x_i^1, x_i^2, \cdots, x_i^j, \cdots x_i^M\}$ denotes the set of modalities of the samples. We want to measure the ITE of modality $j$ in sample $i$. The modality of the treatment variable is denoted as $x_i^j$, the control group is $S(x_i)$, and the treatment group is $S(x_i)\backslash x_i^j$ then its individual causal effect can be expressed as follows:

$$
\begin{aligned}
ITE(x_i^j) = \phi_H[x_i^j] &= \mathcal{V}(H(T)) - \mathcal{V}(H(T')) \\
&= \mathcal{V}(H(S(x_i))) - \mathcal{V}(H(S(x_i)\backslash x_i^j)) \\
&= B(\hat{F}(S(x_i))) - B(\hat{F}(S(x_i)\backslash x_i^j)) \\
&= B(\hat{F}(S(x_i))) - B(\hat{F}(S(x_i)|do(t_i = x_i^j))),
\end{aligned}
\tag{3}
$$

where $B(.)$ is the benefit function presented in the previous subsection. $\hat{F}(.)$ denotes is the output function of the multimodal model. When quantifying the contribution of the modality $j$ in the sample $i$, we must not only calculate the effect of the intervention modality $j$ on the output but also consider its own impact on the output. Thus, its contribution can be expressed as follows:

$$
\begin{aligned}
\Phi(x_i^j) &= ITE(x_i^j) + \mathcal{V}(H(x_i^j)) \\
&= B(\hat{F}(S(x_i))) - B(\hat{F}(S(x_i)|do(t_i = x_i^j))) + B(\hat{F}(x_i^j)).
\end{aligned}
\tag{4}
$$

Additionally, as specified in Eq.1, when the multimodal model makes an incorrect prediction, the control group is $B(\hat{F}(S(x_i)))$=0. The minimum of treatment group $B(\hat{F}(S(x_i)|do(t_i = x_i^j)))$ is 0, and $B(\hat{F}(x_i^j)) \leq 1$. Then we have $\Phi(x_i^j) \leq 1$, at which point we consider this treatment is not positive and the modality $j$ in sample $i$ is weak modality. We conducted further discussions on $ITE$, at Appendix A.6.

### 3.4 Dynamic modality optimization strategy

After quantifying the modality contribution at the sample level, we obtain the contribution of different modalities for each sample, and then distinguish the weaker modalities. We propose Algorithm 1, which implements a dynamic enhancement strategy to optimize weaker modalities during training iterations. To alleviate modality imbalance problem, we design a specific optimization function Eq.5 to selectively enhance the samples of weak modalities. The enhancement of weak modalities inevitably induces augmented modality-specific data within samples, thereby increasing the likelihood of model overfitting during training. We employ adaptive masking to mitigate model overfitting. We mask localized features of specific modalities, thereby compelling the model to leverage cross-modal contextual dependencies to achieve optimization objectives. This method strengthens inter-modal semantic coherence while preventing over-reliance on partial characteristics of individual modalities.

**Algorithm 1** : CModB Algorithm

---

**Input:** Dataset $\mathcal{D} = \{(x_i^m, y_i') | i \in \{1, \ldots, N'\}, m \in \{1, \ldots, M\}\}$(train / val / test), device, method.
**Output:** Learned parameters $\theta$ of multimodal model.
**INIT:** Optimized dataset $D^{op}$, number of modalities $M$, initial parameters $\theta^0$, maximum iterations $E$, learning rate $\eta$, freeze_train epoch $F$.

1: **for** $e = 1$ **to** $E$ **do**
2:     **if** $e < F$ **then**
3:         Update model parameters $\theta$ with dataset $\mathcal{D}$;
4:     **else**
5:         Update $\mathcal{D}^{op} := \mathcal{D}$;
6:         **for** each sample $(x_i^m, y_i')$ in $D$ **do**
7:             Encoder feature extraction $h(\cdot)$;
8:             Feature fusion $f(\cdot)$;
9:             calculate the each unimodal **ITE** using Eq.3;
10:             calculate the each unimodal contribution $\Phi(x_i^j)$ using using Eq.4;
11:             Obtain optimize status using Eq.5;
12:             Update dataset $\mathcal{D}$ by the optimize status to obtain an optimized dataset $\mathcal{D}^{op}$;
13:         **end for**
14:         Update model parameters $\theta$ with dataset $D^{op}$.
15:     **end if**
16: **end for**

---

By applying masking, we can simulate real-world scenarios where data may exhibit incompleteness or partial information loss. We employ Time Masking for audio modality and Spatial Masking for video modality. This approach enhances training data diversity while mitigating the model's sensitivity to localized noise. So that the multimodal model learns to deal with incomplete data during training, and then has better adaptability and robustness when facing various complex situations in the real world.

$$\text{Re}(x^i) = \begin{cases} f_{\text{Re}}(1 - \Phi(x_i^j)) \cdot \text{Mask}(\Phi(x_i^j)) & \Phi(x_i^j) \leq 1, \\ 0 & \Phi(x_i^j) > 1, e <= E/2, \\ \alpha \cdot \text{Mask}(1 - \Phi(x_i^j)) & \Phi(x_i^j) > 1, e > E/2. \end{cases} \tag{5}$$

Here, $f_{\text{Re}}(.)$ is a monotonically increasing function and $\text{Mask}(\cdot)$ is a mask function, e is the number of iterations, E is the maximum iterations of the multimodal model, $\alpha$ is a hyperparameter. We mainly focus on optimizing the weaker modalities with low contribution. We maintain the original $\mathcal{D}$ inputs for the other modalities during the first half of multimodal training, while applying masking to these during the latter phase. We employ time masking for audio modality and spatial masking for video modality We achieve directional and targeted modality rebalancing by the above method.

## 4 Experiments

### 4.1 Experimental setup

**Datasets:** We select five public datasets,including CREMA-D[43], Kinetic Sounds[44], UCF-101[45], CMU-MOSEI[46], and NVGesture[47] datasets to validate our proposed method. The description of the complete dataset is provided in Appendix A.1.

**Implementation Details:** Details of the implementation are given in the Appendix A.2.

**Baselines:** We compare a range of baseline methods. These contain traditional MML approaches and fusion methods for balanced multimodal learning, namely, feature concatenation (Concat), prediction summation (Sum), prediction weighting (Weight) [48], MMCosine [49], AGM [50], OGM [17], GBlending [32], PMR [12], MMCooperation [8], Relearning [51], MLA [10], MMPareto [16].

### 4.2 Comparison with imbalanced MML baselines

We conducted comparative experiments on multiple datasets with various modalities and compared with SOTA methods. The main results for all datasets are presented in Table 1, where "CMoB" denotes

our proposed approach. In Table 1, Unimodal-1 denotes training models using only the audio modality for datasets like CREMA-D, Kinetics Sounds, and CMU-MOSEI. For UCF-101 and NVGesture, this configuration corresponds to the RGB modality. Unimodal-2 infers training exclusively on the video modality for CREMA-D, Kinetics Sounds, and CMU-MOSEI. For UCF-101 and NVGesture, it corresponds to the optical flow modality. Unimodal-3 applies to CMU-MOSEI, where models are trained using the text modality, and to NVGesture, where the depth modality is utilized. It is worth noting that **Concat** refers to the baseline method commonly used in multimodal learning for mitigating modality imbalance problem, which employs concatenation fusion with a single multimodal cross-entropy loss function. Based on the results, the following key observations can be drawn: (1) Our proposed CMoB approach demonstrates strong generalization ability by showing excellent consistent performance across diverse heterogeneous multimodal datasets. Compared with PMR and OGM, which are rebalancing methods only applicable to two modalities, our method can handle datasets with more modalities and achieve the best results. (2) Each of the modality rebalancing frameworks significantly outperform the baseline Concat method relying on direct feature concatenation, with consistent performance improvements observed across evaluation metrics. This empirical evidence substantiates the objective existence of multimodal imbalance challenges, necessitating dynamic modality-specific parameter adjustment during multimodal joint training. (3) In the CMU-MOSEI dataset, there is a clear phenomenon that the best unimodal performance (Unimodal-3) surpasses its multimodal learning counterpart and outperforms many well-performing rebalancing methods on other datasets. In CREMA-D and CMU-MOSEI dataset, conventional fusion methods without modality balancing exhibit only minimal performance gains when compared to the performance of the important unimodal (the dominant modality in multimodal training such as Unimodal-1 in CREMA-D, Unimodal-3 in CMU-MOSEI). We will discuss and visualize this in the next subsections.

Table 1: Comparison with state-of-the-art (SOTA) multimodal learning algorithms. Bold and underlined results denote the best and second-best performances, respectively.

| Method | CREMA-D | | KineticsSounds | | UCF-101 | | CMU-MOSEI | | NVGesture | |
|---|---|---|---|---|---|---|---|---|---|---|
| | ACC | F1 | ACC | F1 | ACC | F1 | ACC | F1 | ACC | F1 |
| Unimodal-1 | 61.17 | 60.63 | 55.06 | 54.96 | 78.60 | 77.49 | 71.09 | 41.7 | 78.22 | 78.33 |
| Unimodal-2 | 49.56 | 47.81 | 45.31 | 43.76 | 59.90 | 58.19 | 71.03 | 41.68 | 78.63 | 78.65 |
| Unimodal-3 | - | - | - | - | - | - | 80.58 | 74.57 | 81.54 | 81.83 |
| Concat | 65.5 | 65.07 | 65.63 | 65.28 | 81.8 | 81.21 | 78.99 | 69.40 | 81.33 | 81.47 |
| Sum | 63.44 | 63.12 | 64.97 | 64.72 | 80.21 | 79.42 | 79.10 | 71.15 | 82.99 | 83.05 |
| Weight | 66.53 | 66.41 | 65.33 | 64.89 | 82.65 | 82.19 | 79.94 | 72.31 | 82.42 | 82.57 |
| MMCosine | 67.19 | 67.34 | 67.49 | 67.09 | 82.97 | 82.47 | 80.38 | 73.67 | 81.52 | 81.55 |
| AGM | 71.59 | 72.11 | 66.62 | 65.88 | 81.7 | 80.89 | 79.86 | 71.89 | 82.78 | 82.82 |
| OGM | 67.76 | 68.02 | 67.04 | 66.95 | 82.07 | 81.3 | - | - | - | - |
| GBlending | 71.59 | 71.72 | 68.82 | 66.43 | 85.01 | 84.5 | 79.64 | 73.29 | 82.33 | 82.91 |
| PMR | 67.19 | 67.20 | 67.11 | 66.87 | 81.93 | 81.48 | - | - | - | - |
| MMCooperation | 75.85 | 76.68 | 68.01 | 68.03 | 85.25 | 84.69 | 79.84 | 72.99 | 82.85 | 83.02 |
| Relearning | 71.02 | 71.46 | 65.92 | 65.48 | 82.87 | 82.15 | 78.75 | 70.02 | 82.87 | 82.94 |
| ARL | 74.19 | 74.63 | 68.40 | 68.75 | 85.12 | 84.41 | - | - | - | - |
| MLA | 79.43 | 79.90 | 69.05 | 68.75 | 85.38 | 84.84 | 78.65 | 70.02 | 83.73 | 83.87 |
| MMPareto | 76.87 | 77.35 | **74.55** | **74.21** | 85.3 | 84.89 | 81.18 | 74.64 | 83.82 | **84.24** |
| CMoB | **79.75** | **79.98** | 72.03 | 71.74 | **86.82** | **86.21** | **81.24** | **74.97** | **84.06** | 84.18 |
| | ±0.27% | ±0.38% | ±0.22% | ±0.32% | ±0.27% | ±0.34% | ±0.19% | ±0.26% | ±0.14% | ±0.21% |

## 4.3 Comparison in scarcely informative modality case

In existing modal rebalancing methods, the mainstream view is to regard modalities with poor prediction performance as weak modalities and conduct additional training during the unimodal balancing. In practical applications, certain modalities may contain extremely limited label-relevant information and instead contain more noise. If the multimodal model is forced to learn information from these modalities under such circumstances, it will memorize more noise, thereby leading to negative outcomes. To simulate this scenario, we follow this paper [51] and choose to carry out experiments on the CREMA-D dataset. The video modality in the CREMA-D dataset consists of standardized emotional performances in a controlled laboratory environment, with its features primarily focusing on facial details of the human face. Compared to its audio modality, it inherently contains less label-relevant information. We modified its audio modal by adding extra white Gaussian noise. The experimental results are visualized as shown in Figure 2 and Appendix A.4.

From the visualization results, we can observe that when the available information in the dataset is very limited, the performance of existing methods all declines. As shown in Figure 2 the Concat

method, the obvious cluster overlap in the video modality of the concat method indicates that it may be difficult to distinguish all categories in the high-dimensional space, and the categories in the audio modality are relatively dispersed. Through a direct comparison with the concat method, it can be visually observed that the modality rebalancing method improves the model's discriminability to a certain extent. This further validates the necessity of modality imbalance learning. In the second row of Figure 2 and Figure 5, the comparison of the video modality shows that under such extreme conditions, all methods have drawbacks. Their data points fail to form distinct separated clusters in the low-dimensional space. Our method shows relatively better performance in the video modality. In the audio modality, our method shows that the data points of the same category are more compact, while those of different categories are more separated. This indicates that our method makes it easier to distinguish these emotional categories. This proves that our method can also exhibit good robustness in such extreme scenarios.

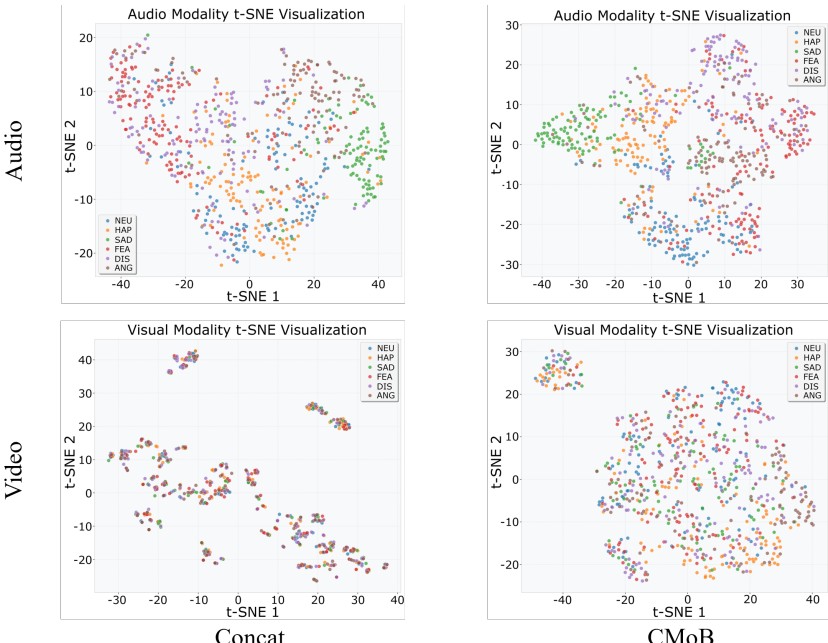

Figure 2: Each Unimodal representation visualization by t-SNE on the processed **CREMA-D** dataset. The six categories are indicated in different colors.

### 4.4 Ablation study

To comprehensively assess the effectiveness of our proposed method, we conduct experiments to study the influence of main components. Here, CQM represents our proposed a causal-aware modality contribution quantification method, and RE denotes the dynamic modality optimization strategy. We conduct an ablation study on CREMA-D and KineticsSounds datasets. The results are shown in Table 2. The implementation of the dynamic modality optimization method (RE) is predicated on the causal-aware modality contribution quantification method (CQM). Consequently, RE is inapplicable in the absence of CQM. From Table 2, we can see that both CQM and RE can boost performance in multimodal learning. Moreover, by integrating CQM with RE, the performance gap between audio modality and video modality is greatly reduced. Ablation studies clearly demonstrate the critical contribution of our proposed modules to overall method performance enhancement.

### 4.5 Further analysis

**Analysis of Modality Gap:** As mentioned in the paper [52], the existence of a geometric phenomenon in multimodal models: Modality Gap. It can denote regions in the shared space where the embeddings of different modalities are significantly separated, and a larger modality gap indicates better performance. We make further comparisons by validating the modality gap between our method and the better performing methods (which outperforms CMoB method with some data). The results are shown in Figure 3. In the concat method, the data points of the audio modality are over-clustered,

Table 2: Results of ablation study on CREMA-D and KineticsSounds datasets

| Dataset | Module | | ACC |
| --- | --- | --- | --- |
| | CQM | RE | |
| CREMA-D | × | × | 65.50 |
| | √ | × | 76.42 |
| | √ | √ | 79.75 |
| KineticsSounds | × | × | 65.63 |
| | √ | × | 70.96 |
| | √ | √ | 72.03 |

while those of the video modality are over-dispersed, and a smaller modality gap. This indicates that the audio modality is primarily leveraged for discrimination in the Concat method. As shown in Figure 3(b) and Figure 3(c) , compared with MLA and MMPareto, our method has a larger modality gap, with more compactly clustered modality representations and fewer outliers, demonstrating its ability to learn more discriminative features and further validating its stability.

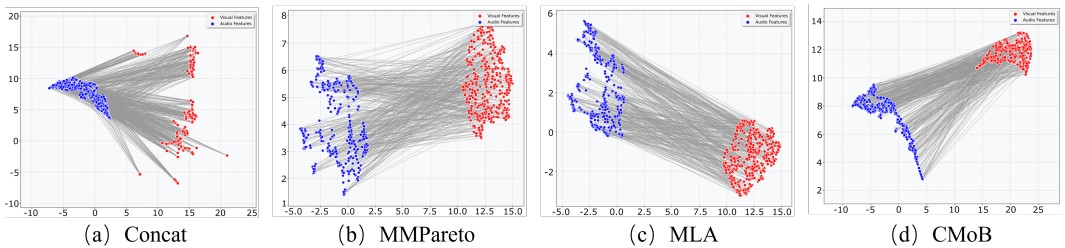

  (a) Concat     (b) MMPareto     (c) MLA      (d) CMoB

Figure 3: Visualizations of the modality gap on CREMA-D dataset.

**Visualization:** We employ Grad-CAM [53] to visualize the key regions focused by various rebalancing methods(Concat, MMPareto, MLA) during training on the CREMA-D dataset. Visualization results and analysis are provided in the Appendix A.3.

## 5 Conclusion

In this paper, we first analyze the limitations in existing modality rebalancing methods that neglect the dynamic variations of modality contributions at the sample level during training. Inspired by the remarkable causal inference power of the human modality choice behavior in cognitive science, we propose a causal-aware modality validation approach for balanced multimodal learning. We employ intervention methods to evaluate the causal effect, quantifying changes in modality contributions at the sample level during multimodal learning. The fine-grained evaluation approach enables targeted optimizations across modalities at the sample level, effectively mitigating the issue of multimodal imbalance. Comparative experimental results in multiple datasets validate the effectiveness of the proposed algorithm.

## Acknowledgments

This work was supported by the National Natural Science Foundation of China (U24A20323 and 62376145), the Key Technologies Program of Taihang Laboratory in Shanxi Province (THYF-JSZX-24010700), the Science and Technology Innovation Talent Team of Shanxi Province (202204051002016), and the Taiyuan City "Double hundred Research action" of the first batch project about "Leading the Charge with Open Competition" (2024TYJB0127).

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

# A   Supplementary Materials and Experimental Details

## A.1   Dataset

CREMA-D is an emotion recognition dataset with two modalities, audio and video. The dataset includes 7,442 clips annotated in six emotions. Kinetic Sounds is an action recognition dataset with two modalities, audio and video. This dataset contains 19k 10-second video clips categorized into 31 human action classes, which are selected from the Kinetics dataset. UCF-101 is an action recognition dataset with two modalities, RGB and optical fow. This dataset contains 10l categories of human actions with 9,537 samples in the training set and 3,783 samples in the test set. CMU-MOSEI is a sentiment analysis dataset with three modalities, audio, video, and text. This dataset includes more than 1,000 online speakers segmented into 23,453 utterances and is annotated with utterance-level sentiment labels. NVGesture is a gesture recognition dataset designed for human-computer interaction research and consists of three modalities, RGB, Depth, and optical flow. This dataset with 1,050 samples in the training set and 482 samples in the test set.

## A.2   The implementation details

In our experiments, we use the raw data for experiments. Following [17, 12, 51], the architecture and initialization setup followed an unbalanced multimodal learning study for a fair comparison. For the CREMA-D and the Kinetic Sounds dataset, ResNet-18 is employed as the backbone for processing both audio and video data and trained from scratch. For the CMU-MOSEI dataset, we employ transformer-based networks as the backbone architecture, training the model from scratch. Encoders used for UCF-101 are ImageNet pre-trained. In term of video and optical flow modalities, we first select 10 frames from each clip and then uniformly sample three frames as input. We adjusted the input channels of ResNet18 from three to one to fit our data format. For audio modal data, we convert to a 257×299 spectrograms for CREMA-D and a 257×1004 spectrograms for KineticsSounds. For text-image datasets, our framework employs ResNet-50 as the image encoder and BERT for text processing, where images are resized to 224×224 resolution and text sequences are truncated to a maximum length of 128 characters. During training, we use the SGD optimizer with momentum (0.9) and set the learning rate at $1 \times 10^{-3}$ . All models are trained on 2 NVIDIA DGX A100.

## A.3   Visualization

We employ Grad-CAM [53] to visualize the key regions focused on by various rebalancing methods(Concat, MMPareto, MLA) during inference on the CREMA-D dataset. By computing gradients of target class scores with respect to the last convolutional feature map and generating pixel-level importance weights, Grad-CAM can precisely localize visual regions that significantly contribute to the decision-making process in the target task (emotion recognition). This visualization method not only helped us to verify the effective use of weak modality features by different models, but also provided an intuitive basis for us to analyze the allocation of visual attention in multimodal interactions. The visualization results are presented in Figure 4, where the first, second, third, fourth and last columns denote the results of the tenth, thirtieth, fiftieth and eightieth last epoch, respectively. This image is a keyframe extracted from the video modality of the category "NEU" in the CREMA-D dataset. Based on the heat map of the video modality, the following key observations can be drawn: (1) In the emotion recognition task, audio modality has richer information than the video modality, especially in the CREMA-D dataset. So we found that the video modality provides limited information at the early stage of training and relies mainly on the audio modality to dominate the feature extraction. At the 50th epoch of training, the video modality of the Concat method starts to focus on the feature extraction of facial expressions. However, the method is optimized by a uniform joint loss function, at this point, the dominant modality (audio modality) leads to a modality imbalance problem that causes the subsequent optimization of the video model to be directionally biased. (2) At different stages of model training, compared to other modal rebalancing methods our method focuses on the features of a person's face that are most relevant to the emotion recognition task, verifying the effectiveness and stability of our method in weak modality learning.

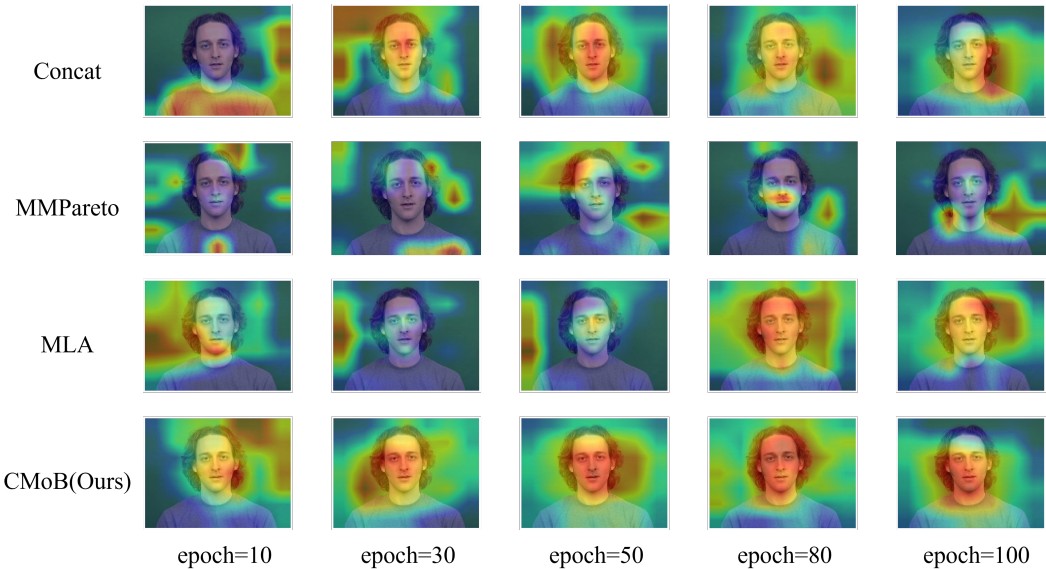

Figure 4: Visualizations of various rebalancing methods on CREMA-D dataset.

## A.4  Comparison in scarcely informative modality case

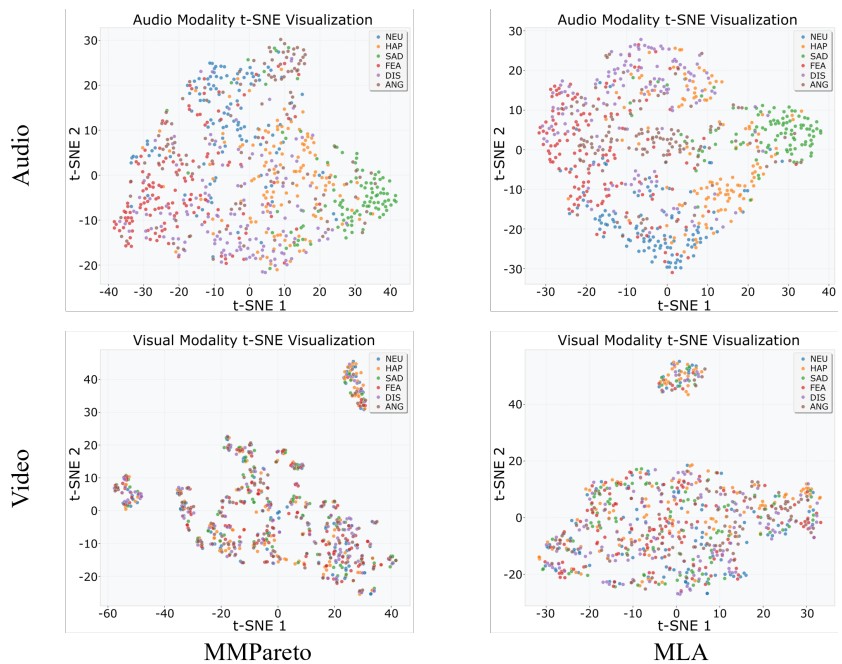

Figure 5: Each Unimodal representation visualization by t-SNE on the processed **CREMA-D** dataset. The six categories are indicated in different colors.

## A.5  Further discussion on ITE

A further question we need to consider is: in multimodal learning, does our evaluation of the individual treatment effect (ITE) possess globality?

"Globality" generally refers to the property that an effect or conclusion holds universally among different individuals, groups, or environments, emphasizing its generalizability and cross-scenario applicability [54, 55]. Take classification tasks as an example: our goal is to train a robust multimodal deep neural network that constructs a generalizable decision boundary, accurately mapping input

data into predefined classes. During training, all multimodal data also operate within the unified deep neural network. After multiple iterations, the resulting discriminative model can classify heterogeneous data from diverse modalities, and it has applicability in different modality in various samples. Therefore, we calculate that the individual causal effect of input modalities and output results processed by the multimodal deep network possesses globality.

## A.6 Comparison with MML baselines on large-scale datasets

We conduct an experimental on the relatively large-scale dataset VGGSound. The VGGSound datasets consist of both audio and video modalities. The VGGSound dataset, which contains 310 classes and a wide range of audio events in everyday life, is a relatively large dataset. It includes 168,618 videos for training and validation, and 13,954 videos for testing. The experimental results show the superiority of our method. The results of the comparative experiment are shown in Table 3

Table 3: Comparison with different methods on VGGSound dataset.

| Method | MAP | ACC |
|---|---|---|
| AGM | 51.98% | 47.11% |
| MLA | 54.73% | 51.65% |
| ReconBoost | 53.87% | 50.97% |
| MMPareto | 54.74% | 51.25% |
| Ours | **54.98%** | **51.74%** |

To further validate the effectiveness of our method under scenarios with more modalities, we follow this paper and conduct further experiments on the Caltech101-20 dataset [8]. We compare our method with the Concat method and the Shapley value method. The experimental results confirm the effectiveness of our method, even when using five modalities (views), as shown in Table 4.

Table 4: Accuracy of our methods on Caltech101-20 dataset.

| Num of modalities | Concat | Shapley | Ours |
|---|---|---|---|
| 2 | 82.91 | 83.47 | **83.71** |
| 3 | 87.71 | 87.99 | **88.22** |
| 4 | 93.64 | 94.07 | **94.35** |
| 5 | 94.63 | 94.73 | **94.86** |

