# OpenReview forum: "CMoB: Modality Valuation via Causal Effect for Balanced Multimodal Learning"
_NeurIPS.cc/2025/Conference — NeurIPS 2025 poster_

### Official Review · Reviewer_DU6H · 2025-06-26

**Clarity:** 3
**Significance:** 3
**Originality:** 3
**Rating:** 4
**Confidence:** 2

**Summary:**

This paper introduces Causal Modality Evaluation (CMoB) to address modality imbalance in multimodal learning. CMoB uses Shannon information uncertainty theory for a gain function, combines causal learning to measure fine - grained modality contributions within samples, and applies dynamic optimization to strengthen weak modalities.

**Questions:**

Can the quantification of causal effects explain the synergistic/competitive relationships between modalities? For example, the complementarity of audio-video modalities in emotion recognition.

**Ethical Concerns:**

["NO or VERY MINOR ethics concerns only"]

**Final Justification:**

The authors' reply addresses most of my concerns. So I keep my initial score.

**Limitations:**

yes

**Quality:**

3

**Strengths And Weaknesses:**

Pros
1. Combining causal learning with information theory to dynamically evaluate modality contributions at the sample level breaks through the limitations of traditional modality - level analysis.
2. The benefit function is constructed based on Shannon information theory, and the causal effect derivation is complete.
3. Compared with more than 10 baseline methods on 5 cross-modal datasets, the results show the best performance in all cases.

Cons
1. The global nature of ITE is mentioned in the paper, but the consistency under different modality combinations is not sufficiently verified.
2. Lacking ablation experiments to verify the effectiveness of each module.
3. The specific implementation of the Mask(·) function in Equation 5 is not explicitly described.
4. It is recommended to add efficiency tests on large-scale datasets and cross-modal number extension experiments.

---

> ### Author Rebuttal · Authors · 2025-07-30
>
> $Q1:$The global nature of ITE is mentioned in the paper, but the consistency under different modality combinations is not sufficiently verified.
>
> $R1:$Thank you for your comment. The "global nature" means our individual treatment effect (ITE) evaluation applies to different modality combinations, but the ITE values for each modality vary across these combinations. We conducted four different modality combinations ：{T+A}, {T+V}, {V+A}, { T+A+V } for experimental verification using three modality datasets (CMU-MOSEI). The Concat method is a baseline method.  The comparative experiments demonstrate the superiority and generalizability of our method, as shown in Table 1 .
>
> Table1 Comparison with different modality combinations on CMU-MOSEI datasets. The evaluation metric is Accuracy (ACC).
> | Dataset | Method  | T+A  | T+V  | V+A  | T+A+V |
> |---------|---------|------|------|------|-------|
> | CMU-  MOSEI  | Concat  | 0.745| 0.751| 0.656| 0.789 |
> |   CMU-  MOSEI | Ours    | 0.796| 0.773| 0.701|0.812|
>
> $Q2:$Lacking ablation experiments to verify the effectiveness of each module.
>
> $R2:$ Thanks for your comment. We conducted an ablation study to demonstrate the effectiveness of the proposed method on CREMA-D and  KineticsSounds datasets , and the results are shown in the Table 2. Here, CQM represents our proposed a causal-aware modality contribution quantification method, and RE denotes the dynamic modality optimization Strategy.  Ablation studies clearly demonstrated the critical contribution of our proposed modules to overall method performance enhancement.
>
> Table2 Ablation analysis of proposed modules.
> | Dataset  | Module    |       | ACC    |       |
> |----------|-----------|-------|--------|-------|
> |          | CQM       | RE    |        |       |
> | CREMA-D  | ×         | ×     | 65.50  |       |
> |          | √         | ×     | 76.42  |       |
> |          | √         | √     | 79.75  |       |
> | KineticsSounds | ×    | ×     | 65.63  |       |
> |          | √         | ×     | 70.96  |       |
> |          | √         | √     | 72.03  |       |
>
> $Q3:$The specific implementation of the Mask(·) function in Equation 5 is not explicitly described.
>
> $R3:$Thank you for your comment. We employed **Time Masking** for audio modality and **Spatial Masking** for video modality.
>
> $Q4:$It is recommended to add efficiency tests on large-scale datasets and cross-modal number extension experiments.
>
> $R4:$Following your suggestion, we conduct an experimental on large-scale dataset VGGSound. The VGGSound datasets consist of both audio and video modalities. The VGGSound dataset, which contains 310 classes and a wide range of audio events in everyday life, is a relatively large dataset. It includes 168,618 videos for training and validation, and 13,954 videos for testing.The experimental results show the superiority of our method. The results of the comparative experiment are shown in Table 3.
>
>
> Table3 Comparison with different methods on VGGSound dataset.
> | Method      | MAP     | ACC     |
> |-------------|---------|---------|
> | AGM         | 51.98%  | 47.11%  |
> | MLA         | 54.73%  | 51.65%  |
> | ReconBoost  | 53.87%  | 50.97%  |
> | MMPareto    | 54.74%  | 51.25%  |
> | Ours        | 54.98%  | 51.74%  |
>
> In the manuscript we have already submitted, we have conducted experiments on more-than-two modalities datasets CMU-MOSEI and NVGesture. To further validate the effectiveness of our method under scenarios with more modalities, we follow this paper and conduct further experiments on the Caltech101-20 dataset [1]. We compare our method with the Concat method and the Shapley  value method. The experimental results demonstrate the effectiveness of our method, as shown in Table 4.
>
> Table4 Accuracy of our methods on Caltech101-20 dataset.
> | Num of modalities | Concat | Shapley | Ours   |
> |-------------------|--------|---------|--------|
> | 2                 | 82.91  | 83.47   | 83.71  |
> | 3                 | 87.71  | 87.99   | 88.22  |
> | 4                 | 93.64  | 94.07   | 94.35  |
> | 5                 | 94.63  | 94.73   | 94.86  |
>
> $Q5:$Can the quantification of causal effects explain the synergistic/competitive relationships between modalities? For example, the complementarity of audio-video modalities in emotion recognition.
>
> $R5:$Thank you for your question. We can indeed explain synergistic/ competitive relationships between modalities by quantifying causal effects of each modality. We treat each modality as a treatment variable and the emotion label as the outcome. Using our defined benefit function B(M), we compute the causal effect of each modality. For example, when processing with audio–video modalities in emotion recognition:
>
> • **Synergistic**: During iterative training, when a single modality cannot predict the emotion accurately, but adding the other modality achieves correct prediction, we have $ ITE(audio/video) $=$
>  B\bigl(\hat{F}\bigl(S(x)\bigr)\bigr) - B\Bigl(\hat{F}\bigl(S(x) \mid do(t = x^{(video/audio)})\bigr)\Bigr)$ = $B(multimodal) - B(video/audio)$ = $2$, indicating a synergistic relationship where audio and video are complementary.
>
> • **Competitive**: During iterative training, if the audio modality alone successfully predicts the emotion label, yet adding the video modality yields an accurate prediction with lower confidence than that achieved using the audio modality alone. At this point, $ITE(video)$=$
>  B\bigl(\hat{F}\bigl(S(x)\bigr)\bigr) - B\Bigl(\hat{F}\bigl(S(x) \mid do(t = x^{(audio)})\bigr)\Bigr)
> $
> =$-1$, and the relationship between the two modalities is competitive.
>
> When the causal effect of a modality is less than 1, it negatively impacts emotion prediction; if it is greater than 1, the impact is positive.
>
> [1]Wei Y, Feng R, Wang Z, et al. Enhancing multimodal cooperation via sample-level modality valuation[C]. Proceedings of the Conference on Computer Vision and Pattern Recognition. 2024: 27338-27347.

---

> > ### Comment · Reviewer_DU6H · 2025-08-07
> >
> > Thank the authors for the detailed reply! It addresses most of my concerns. I will keep my initial score.

---

### Official Review · Reviewer_iCFM · 2025-06-30

**Clarity:** 3
**Significance:** 3
**Originality:** 3
**Rating:** 4
**Confidence:** 4

**Summary:**

This paper proposes the CMoB framework to address the "modal imbalance" problem in multimodal learning: first use the entropy-inspired sample benefit function to measure the confidence gain of each sample after adding different modalities, then treat each modality as a "process" to calculate the individual treatment effect (ITE), and finally dynamically shield and enhance the weak modality. Experiments on five datasets show that CMoB generally outperforms or approaches the latest balancing methods in terms of accuracy/F1, and its improvement effect on weak modalities is verified by t-SNE and Grad-CAM analysis.

**Questions:**

* How do the benefit‐function values and the final performance figures change if temperature-scaled logits or negative log-likelihood are used instead of the default soft-max confidence?
* For each benchmark, what is the percentage increase in wall-clock training time and peak GPU memory consumption relative to the strongest baseline?
* Please report an ablation study that quantifies the individual gains contributed by each module of the proposed framework.

**Ethical Concerns:**

["NO or VERY MINOR ethics concerns only"]

**Final Justification:**

The authors’ response has satisfactorily addressed my concerns. I keep my positive score.

**Limitations:**

yes

**Paper Formatting Concerns:**

none.

**Quality:**

3

**Strengths And Weaknesses:**

**Strength:**
* This work is the first to incorporate causal Individual Treatment Effect (ITE) estimation into sample-level modality valuation, enabling a fine-grained quantification of each modality’s contribution. The introduction of causal inference (ITE estimation) into the modality rebalancing task is novel and insightful. It enriches the modality contribution evaluation beyond simple gradient or attention-based metrics.
* The shift from global, modality-level contribution estimation to per-sample analysis allows CMoB to dynamically capture fine-grained imbalance and perform more targeted optimizations.
* Extensive experiments across multiple benchmarks substantiate the effectiveness and robustness of the proposed approach.

**Weaknesses:**
* Per-sample ablation and dynamic masking are likely to incur substantial additional training time and computational overhead.
* The paper lacks an ablation study that disentangles the respective contributions of the benefit function, the ITE computation, and the dynamic masking mechanism.
* It is not recommended to share code via GitHub during the review process, as it may compromise the double-blind review principle. Instead, it is advisable to use an anonymous code-sharing platform such as https://anonymous.4open.science/.
process, as it may compromise the double-blind review principle. Instead, it is advisable to use an anonymous code-sharing platform such as https://anonymous.4open.science/.
* Typo: "modality-spcifice" should be "modality-specific" in line 157.

---

> ### Author Rebuttal · Authors · 2025-07-30
>
> $Q1:$How do the final performance figures change if temperature-scaled logits or negative log-likelihood are used instead of the default soft-max confidence?
>
> $R1:$Following your suggestion, we replaced the default softmax confidence with temperature-scaled logits. We set the temperature parameter T=2 and T=0.5. The evaluation metric is Accuracy (ACC). The experimental results demonstrate that using temperature-scaled logits(T=2) is helpful for the target task. In our method, we use cross-entropy loss, which is essentially softmax + negative log-likelihood. Our method(CMoB) isn't restricted to using cross - entropy loss and softmax confidence. We mainly use them for a fair comparison with other methods.
>
> | Method                | CREMA-D | KineticsSounds |
> |---------------------|---------|-------------|
> | CMoB(softmax)            | 79.75    | 72.03    |
> | CMoB(temperature-scaled logits(T=2)) | 79.92   | 72.41   |
> | CMoB(temperature-scaled logits(T=0.5)) | 79.04   | 71.88   |
>
> $Q2:$For each benchmark, what is the percentage increase in wall-clock training time and peak GPU memory consumption relative to the strongest baseline?
>
> $R2:$Thanks for your comment. Our modality selection and dynamic masking method incur higher computational overhead. We compare training time and computational overhead with the baseline model that performs best. Among them, the MLA method is the strongest baseline in the CREMA-D dataset, and the MMPareto method is the strongest baseline in the KineticsSounds dataset. Preliminary results show that our method increases GPU memory consumption by approximately 20% to35%. We consider that this overhead is acceptable given the improvement in the accuracy of the experimental results, as shown in the table below. Compared with methods that use data processing strategies (e.g., Shapley value) to address  modality imbalance, our method achieves lower computational overhead and reduced training time.
>
> | Method    | CREMA-D             |         |       | CMU-MOSEI       |       |         |
> |-----------|---------------------|----------------|----------------|-----------------|----------------|----------------|
> |           | training time       | GPU memory | ACC    | training time   | GPU memory   | ACC   |
> | MMPareto  | 4h35min      | 10672MiB | 76.87     | 7h29min         | 15598MiB  |  81.18   |
> | MLA       | 4h14min            | 9295MiB     |  79.43 | 6h53min         | 13841MiB       |78.65|
> | Shapley value  | 5h52min            | 13514MiB |     77.82 | 9h41min         | 24384MiB       |79.87|
> | CMoB      | 5h37min            | 13014MiB  |   79.75  | 8h34min         | 19030MiB       |81.24|
>
> $Q3:$Please report an ablation study that quantifies the individual gains contributed by each module of the proposed framework.
>
> $R3:$Thanks for your comment. We conduct an ablation study to demonstrate the effectiveness of the proposed method on CREMA-D and KineticsSounds datasets, and the results are shown in the table below. Here, CQM represents our proposed a causal-aware modality contribution quantification method, and RE denotes the dynamic modality optimization Strategy. Ablation studies clearly demonstrate the critical contribution of our proposed modules to overall method performance enhancement.
>
> | Dataset  | Module    |       | ACC    |       |
> |----------|-----------|-------|--------|-------|
> |          | CQM       | RE    |        |       |
> | CREMA-D  | ×         | ×     | 65.50  |       |
> |          | √         | ×     | 76.42  |       |
> |          | √         | √     | 79.75  |       |
> | KineticsSounds | ×    | ×     | 65.63  |       |
> |          | √         | ×     | 70.96  |       |
> |          | √         | √     | 72.03  |       |

---

> > ### Comment · Reviewer_iCFM · 2025-08-05
> > **Comments**
> >
> > I thank the authors for their rebuttal. My concerns have been resolved, and I maintain my positive score.

---

### Official Review · Reviewer_WgAE · 2025-07-02

**Clarity:** 2
**Significance:** 2
**Originality:** 2
**Rating:** 4
**Confidence:** 5

**Summary:**

The paper introduces a causal-aware modality valuation approach (CMoB) for addressing the problem of modality imbalance in multimodal learning. This method allows for the balancing of modality contributions at the sample level, helping to improve the performance of weak modalities while mitigating modality imbalance. The paper demonstrates the effectiveness of CMoB through experiments on several multimodal datasets.

**Questions:**

Please refer to the above section.

**Ethical Concerns:**

["NO or VERY MINOR ethics concerns only"]

**Final Justification:**

After all the authors' responses, many of my concerns are addressed. Hence, I raise my score.

**Limitations:**

Please refer to the above section.

**Paper Formatting Concerns:**

No formatting concerns.

**Quality:**

2

**Strengths And Weaknesses:**

Strength:

1. The method focuses an important issue in multimodal learning, modality imbalance, and provides a potential solution to improve the integration of diverse modalities in complex datasets.
2. The paper validates its approach with experiments on multiple multimodal datasets, showcasing the effectiveness of the proposed method.

Weakness:

1. The motivation behind the paper is not clear. The authors mention being inspired by the human nervous system but then shift to a causal learning perspective without clearly explaining the connection between these two concepts. The introduction section lacks logical coherence in presenting the proposed approach.
2. The proposed method claims to solve modality imbalance from a causal learning perspective. However, the actual operation involves comparing the effects of including or excluding a modality, which is conceptually similar to existing methods like Shapley value. What are advantages of this approach over Shapley value or other similar techniques?
3. The paper's writing style needs improvement, especially in the introduction, which spends excessive time discussing related work but fails to clearly explain the motivation behind the proposed method. In addition, Section 3.1 is too detailed and spends considerable space describing concepts like cross-entropy, which do not directly contribute to the core argument of the paper. This section could be more concise and focused.
4. In Section 4.3, the authors claim to replicate the experiment from [45] on the "scarcely informative modality" case but using the original CREMA-D dataset. However, [45] created such a case by adding significant noise to the audio modality of the CREMA-D dataset. The original CREMA-D dataset does not inherently contain "scarcely informative modality" cases.
5. The quality of the images in the paper, particularly Figures 2 and 3, is poor. The legends in these figures are difficult to read, which diminishes the clarity of the data presented.

---

> ### Author Rebuttal · Authors · 2025-07-30
>
> $Q1:$The authors mention being inspired by the human nervous system but then shift to a causal learning perspective without clearly explaining the connection between these two concepts.
>
> $R1:$ Thanks for your comment. Through analyzing the human cognitive system's processing of multimodal data, we observe that it first extracts discriminative causal features from raw multimodal inputs via experientially-guided mechanisms. Then, by using information entropy to dynamically quantify causal contributions across modalities, this process corresponds to the principle of "**Causal  Effect**"in causal learning. Therefore, our framework simulates the cognitive system's " Dynamic Evaluation of Modality Contribution " by quantifying the causal effects at the sample level .
>
> $Q2:$What are advantages of this approach over Shapley value or other similar techniques?
>
> $R2:$Thanks for your comment. Shapley value method inevitably leads to exponentially longer training times and heavier computational burdens. Our method (CMoB) reduces this burden substantially, while providing theoretical guarantees through causal intervention principles. For example, when evaluating the contribution of the $M^3$ modality in a dataset containing three modalities {$M^1$, $M^2$,$M^3$}, the Shapley value method is necessary to calculate the combinations of four subsets {∅, $M^1$, $M^2$, $M^1$+$M^2$}. Our causal learning framework only calculates the {$M^1$+$M^2$} coalition. The peak GPU memory consumption on the CMU-MOSEI dataset was 19030MiB for CMOB method and 24384MiB for Shapley value method . Comparison experiments with the baseline of Shapley values[1] further demonstrate the superiority of our method, as shown in the table below.
>
> Table1 Comparison with Shapley value algorithms. The evaluation metric is Accuracy (ACC).
> | Method   | CREMA-D | KineticsSounds | UCF-101 | CMU-MOSEI | NVGesture |
> |---------|---------|-----------|---------|---------|----------|
> | Concat  | 65.5     | 65.63    | 81.8   | 78.99   | 81.33   |
> | Shapley value | 77.82   | 68.01    | 85.25  | 79.87   | 82.87  |
> | Ours   | 79.75   | 72.03    | 86.82  | 81.24   | 84.06    |
>
> $Q3:$The paper's writing style needs improvement, especially in the introduction, which spends excessive time discussing related work but fails to clearly explain the motivation behind the proposed method. In addition, Section 3.1 is too detailed and spends considerable space describing concepts like cross-entropy, which do not directly contribute to the core argument of the paper. This section could be more concise and focused.
>
> $R3:$Thanks for your question. To address modality imbalance problem, current methods primarily rely on gradient variations to measure modality contribution. These methods primarily rely on modality-level contribution assessments to measure gaps in representational capabilities and enhance poorly learned modalities, overlooking the dynamic variations of modality contributions across individual samples. Although Shapley value-based methods can measure modality contributions at the sample level, they cause a sharp increase in computational overhead and training time on datasets with more than two modalities. Inspired by human cognitive science, we propose a causal-aware modality contribution quantification method from a causal perspective to capture fine-grained changes in modality contribution degrees within samples. And we dynamically select and optimize modalities based on real-time changes in their contributions.
>
> In addition, In Section 3.1, we primarily formalize the modality imbalance problem. We will simplify section 3.1 and emphasize only components essential to modality imbalance understanding.
>
> $Q4:$ In Section 4.3, the authors claim to replicate the experiment from [45] on the "scarcely informative modality" case but using the original CREMA-D dataset. However, [45] created such a case by adding significant noise to the audio modality of the CREMA-D dataset. The original CREMA-D dataset does not inherently contain "scarcely informative modality" cases.
>
> $R4:$Thanks for your comment. Here we may not have provided a clear expression. We modify the audio data of the CREMA-D dataset, adding extra white Gaussian noise to make it noisier and scarcely discriminative. In this section, we conduct comparative experiments using the **processed** CREMA-D dataset.
>
> $Q5$:The quality of the images in the paper, particularly Figures 2 and 3, is poor. The legends in these figures are difficult to read, which diminishes the clarity of the data presented.
>
> $R5:$Thanks for your comment. We acknowledge that the current presentation does not adequately convey critical details. The legends in these figures are difficult to decipher because the annotation elements were not scaled appropriately. In the next version, we will use a higher resolution (600 dpi) and adjust the font size of all annotations to 10pt+, and increase the line thickness by 50% in Figures 2 and 3.
>
> [1]Wei Y, Feng R, Wang Z, et al. Enhancing multimodal cooperation via sample-level modality valuation[C]. Proceedings of the Conference on Computer Vision and Pattern Recognition. 2024: 27338-27347.

---

> > ### Comment · Reviewer_WgAE · 2025-08-05
> >
> > Thank you for the authors' response, which has addressed some of my concerns. However, as per the current version, I am still unconvinced of the correlation between the proposed method and cognitive science. It seems unnecessary to introduce cognitive science here, as it only adds confusion.
> >
> > One more question: it seems that there is no accuracy comparison reported in the experiments for the scarcely informative modality case.
> >
> > I prefer to keep my original score, and strongly recommend refining the paper thoroughly, particularly the motivation and method descriptions.

---

> > > ### Author Response · Authors · 2025-08-07
> > > **We conducted supplementary experiments in response to the reviewers' questions. We further elaborated on the reviewers' questions regarding the correlation between the proposed method and cognitive science.**
> > >
> > > $Q1:$ **It seems that there is no accuracy comparison reported in the experiments for the scarcely informative modality case.**
> > >
> > > $R1:$ We sincerely thank you for your suggestion. We conducted a comparison experiment to demonstrate the effectiveness of the proposed method on the **processed** CREMA-D dataset (scarcely informative modality case).  The experimental results demonstrate the effectiveness of our method, as shown in Table 1.
> > >
> > > Table1 Comparison with imbalanced multimodal learning methods on the **processed** CREMA-D dataset(scarcely informative modality case).
> > > | Method    | ACC | F1  |
> > > |-----------|-------------------------------------|------------------------------------|
> > > | OGM       | 62.63                               | 65.07                              |
> > > | Greedy    | 63.17                               | 63.83                              |
> > > | PMR       | 65.73                               | 65.33                              |
> > > | AGM       | 62.87                               | 63.73                              |
> > > | Relearning| 67.49                               | 68.28                              |
> > > | MLA       | 68.34                               | 69.81                              |
> > > | Ours      | 68.75                               | 69.88                             |
> > >
> > > $Q2:$**The correlation between the proposed method and cognitive science.**
> > >
> > > $R2: $ The focus of our manuscript centers on the challenge of modality imbalance in multimodal learning. Existing research in cognitive science [1-4] demonstrates that when the number of modalities within a sample space increases, the human brain can **dynamically assess** and extract discriminative feature information from heterogeneous multimodal data, continuously refining **Cognitive Processes** through **adaptive learning**. This aligns with the two key properties summarized in our manuscript: **1)** Modality Contribution Valuation, and  **2)** Granular Adjustment at the Sample Level. Our research aims to establish a multimodal learning framework that computationally formalizes this **Cognitive Processes**, thereby mitigating the modality imbalance problem.
> > >
> > > We reiterate our appreciation for your question regarding ''the connection between cognitive science and our proposed method." This inquiry has deepened our reflection on the rationality of the correlation between the two. We will provide a more explicit articulation of their relationship in the revised version.
> > >
> > > Should our interpretation not fully address your concerns, we remain open to further scholarly discourse to contribute to this field of study.
> > >
> > > [1] Silveira I, Varandas R, Gamboa H. Cognitive lab: A dataset of biosignals and HCI features for cognitive process investigation[J]. Computer Methods and Programs in Biomedicine, 2025, 269: 108863.
> > >
> > > [2] Shen X, Hu X, Zhang R, et al. A Lightweight Triple-Modal Fusion Network for Progressive Mild Cognitive Impairment Prediction in Alzheimer's Disease[J]. Frontiers in Neuroscience, 2025, 19: 1637291.
> > >
> > > [3] Occhipinti A , Verma S , Doan T A C .Mechanism-aware and multimodal AI: beyond model-agnostic interpretation[J].Trends in Cell Biology, 2024, 34(2):85-89.
> > >
> > > [4]Xuelong L. Multi-Modal Cognitive Computing[J]. SCIENTIA SINICA Informationis, 2022, 53(1):1-32.

---

### Official Review · Reviewer_2axZ · 2025-07-03

**Clarity:** 2
**Significance:** 3
**Originality:** 2
**Rating:** 4
**Confidence:** 4

**Summary:**

The authors discuss the limitations in existing modality rebalancing methods and point out they neglect the dynamic variations of modality contributions at the sample level during training. They propose a causal-aware modality validation approach for balanced multimodal learning. An intervention method is introduced to evaluate the causal effect, quantifying changes in modality contributions at the sample level. The fine-grained evaluation approach enables targeted optimizations across modalities at the sample level, effectively mitigating the issue of multimodal imbalance. Experimental results on multiple datasets show the effectiveness of the proposed algorithm.

**Questions:**

The authors should compare their work with some relatively new algorithms.

**Ethical Concerns:**

["NO or VERY MINOR ethics concerns only"]

**Final Justification:**

The authors have compared the proposed method with some recent methods, validating their effectiveness. I have accordingly raised my rating to 4. But the writing quality should be further improved.

**Quality:**

2

**Strengths And Weaknesses:**

Strengths:
The authors propose a causal-aware modality valuation method to evaluate the sample-level modality contribution in multimodal training.
The authors design an optimization strategy for modality selection at the sample-level according to the contribution degree of each modality in order to mitigate the modality imbalance problem.
The authors validate the effectiveness of the proposed method by comparison and ablation experiments on publicly available data.

Weaknesses:
1. Some important references are missing. Some related works about sample-level modality valuation and balanced learning, such as PDF (Cao, et al., ICML 2024), SMLS (Zhou, et al., Information Fusion 2025), and ARL (Wei, et al., ICML 2025), should be discussed in the related works.

2. The authors should compare one or some relatively new algorithms, such as “Zhou, et al.: Dataset-aware Utopia modality contribution for imbalanced multimodal learning. Information Fusion 2025,” “Ma, et al.: Improving Multimodal Learning Balance and Sufficiency through Data Remixing. ICML 2025,” and “Zhou, et al.: Sample-level Self-paced Learning to Tackle Multimodal Imbalance Problem. ICASSP 2025.”

3. The language quality of the submission is not very good. Some presentation issues should be addressed:
(1) For example, “We” should be “we” in Line 143.
(2) “we define a benefit function” should be “We define a benefit function” in Line 8 of the Abstract.
(3) “we addresses the above problem” should be “we address the above problem” in Line 69 and Line 70.
(4) “modality-specifice” should be “modality-specific,” which is used many times in this paper, such as in Line 157, Line 161, and Line 170.

In addition to the above issues, there are some confusing descriptions, such as “modality data”, I guess it should be “modality-specific data.” I truly suggest the authors carefully check the whole paper and improve the overall presentation quality.

---

> ### Author Rebuttal · Authors · 2025-07-30
>
> $Q1:$ Some important references are missing. The sample-level modality contribution was discussed in "Cao, et al: Predictive Dynamic Fusion. ICML 2024".
>
> $R1:$ Thanks for your comment. Although both our method and the PDF[1] method are sample-level modality contribution quantification, the two methods focus on different aspects. Our method(CMoB), using causal learning and intervention-based contribution estimation, offers better interpretability. CMoB focuses on identifying dominant/ weak modalities at the sample-level during training, whereas PDF focuses on assessing quality disparities among modalities. CMOB emphasizes enhancing learning for weak modalities, whereas PDF aims to reduce reliance on low-quality modalities via Relative Calibration (RC). We conduct comparative experiments with the PDF method  on the CREMA-D dataset. The experimental results show that our proposed method exhibits strong robustness, as shown in the Table1. We will discuss Cao et al (ICML 2024) in the Related Work section and include experimental comparisons in the revision.
>
> $Q2:$ The authors should compare some relatively new algorithms.
>
> $R2:$ Following your suggestion, we conduct comparative experiments with the latest modality imbalance algorithms on the CREMA-D dataset, as shown in the Table 1. To further validate our approach, we add Gaussian noise on 50% modalities and ϵ presents the noise degree. The experimental results show the superiority of our method. In our manuscript , we also compared with relatively new algorithms, such as the MMPareto, MLA, Relearning, and MMCooperation methods were proposed in 2024, and the MMCosine, PMR, and AGM methods were proposed in 2023.
>
> Table1 Comparison with the latest algorithms on the CREMA-D dataset. The evaluation metric is Accuracy (ACC).
> | Method        | ε = 0.0       | ε = 5.0       |
> |--------------|---------------|-------------|
> | Concat        | 61.56±1.37     | 52.33±3.32    |
> | LATE FUSION    | 61.81±2.13     | 49.84±3.72    |
> | PDF[1]         | 63.31±1.11     | 57.85±2.04    |
> | Shapley[2]     | 77.82         | 70.44        |
> | OPM[3]     | 68.45        | 64.25        |
> | ERL-MR[4]     | 74.91      | 69.53        |
> | Utopia[5]     | 78.67      | 68.89        |
> | Ours(CMoB)    | 79.75       | 70.72        |
>
> $Q3:$ The language quality of the submission is not very good.
>
> $R3:$  Thanks for your comment. We will comprehensively revise the manuscript to improve its clarity, grammatical accuracy, and conciseness in the next version.
>
> [1] Cao B, Xia Y, Ding Y, et al. Predictive Dynamic Fusion[C].Proceedings of the International Conference on Machine Learning. PMLR, 2024: 5608-5628.
>
> [2] Wei Y, Feng R, Wang Z, et al. Enhancing multimodal cooperation via sample-level modality valuation[C]. Proceedings of the Conference on Computer Vision and Pattern Recognition. 2024: 27338-27347.
>
> [3]Wei Y, Hu D, Du H, et al. On-the-fly modulation for balanced multimodal learning[J]. IEEE Transactions on Pattern Analysis and Machine Intelligence, 2025:47(1):469–485.
>
> [4]Han W, Cai C, Guo Y, et al. Erl-mr: Harnessing the power of euler feature representations for balanced multi-modal learning[C]. Proceedings of the 32nd ACM International Conference on Multimedia. 2024: 4591-4600.
>
> [5]Zhou Y, Liang X, Xu Y, et al. Dataset-aware Utopia modality contribution for imbalanced multimodal learning[J]. Information Fusion, 2025: 103383.

---

> > ### Comment · Reviewer_2axZ · 2025-08-06
> > **Thanks for the response.**
> >
> > Thanks for the response. The newly added results addressed my concerns, and I decided to raise my rating to 4 (borderline accept). But I still suggest the authors carefully check this paper and improve the overall quality thoroughly.

---

### Note · Authors · 2025-08-13

Dear Area Chairs and Reviewers,

We sincerely thank you for the comprehensive review of our work and the valuable feedback provided. The focus of our manuscript centers on the challenge of modality imbalance in multimodal learning.  We propose a causal-aware modality contribution quantification method from a causal perspective to capture fine-grained changes in modality contribution degrees within samples, thereby mitigating the modality imbalance problem. All four Reviewers have affirmed the significance of our research problem. Specifically, Reviewer iCFM explicitly commending our introduction of causal inference (ITE estimation) into the modality rebalancing task as novel and insightful.

We have addressed all Reviewers' requests for additional experiments, which has received unanimous approval.

We note that Reviewer WgAE questioned the correlation between the our methodology and cognitive science.

 We have addressed this concern in detail in our rebuttal. Here, we provide further clarification.
Existing research in **Cognitive Science** [1,2] (PNAS, Nature) demonstrates that when the number of modalities within a sample space increases, the human brain can **dynamically assess** and extract discriminative feature information from heterogeneous multimodal data, continuously **refining Cognitive Processes through adaptive learning**. This aligns with the two core elements of our method: 1) Modality Contribution Valuation, and 2) Granular Adjustment at the Sample Level. Our research aims to establish a multimodal learning framework that **computationally formalizes this Cognitive Processes** to mitigate the modality imbalance problem.

We hope these clarifications help convey the value of our contributions. We will integrate reviewer suggestions and refine relevant sections in the revised manuscript.

We once again thank all reviewers and the committee for their time, effort, and thoughtful feedback, and we look forward to your final decision.

[1]Rideaux R, Storrs K R, Maiello G, et al. How multisensory neurons solve causal inference[J]. **Proceedings of the National Academy of Sciences**, 2021, 118(32): e2106235118.

[2] Khilkevich A, Lohse M, Low R, et al. Brain-wide dynamics linking sensation to action during decision-making[J]. **Nature**, 2024, 634(8035): 890-900.

Sincerely,

The Authors

---

### Decision · Program_Chairs · 2025-09-17

**Decision:**

Accept (poster)

**Comment:**

The paper tackles the problem of modality imbalance in multimodal learning paradigm via causality-aware modality contribution quantification to capture more granular variations in modality contribution degrees in samples.
Some notable positive aspects mentioned by the reviewer are:
- the paper addresses an important issue in multimodal learning with causality perspective
- first work to introduce causal individual treatment effect
- Extensive experimental results show effectiveness and robustness

Some concerns identified by the reviewers were:
- how different is to Shapley value techniques
- the motivation is not clear
- substantial cost due to per sample ablation
- improvements in writing at some places
- missing references to some relevant related works

After the post-rebuttal discussion period, the reviewers acknowledged that their major concerns have been resolved. For example, the comparison with latest methods, computational cost overhead compared to other aspects, cognitive science aspect, and new experimental results on accuracy comparison have been resolved. Therefore, the AC decides to recommend the acceptance of the paper and recommend authors to incorporate important reviewer's comments in the final version.